# Asparagine614 Determines the Transport and Function of the Murine Anti-Aging Protein Klotho

**DOI:** 10.3390/cells13201743

**Published:** 2024-10-21

**Authors:** Zahra Fanaei-Kahrani, Christoph Kaether

**Affiliations:** Leibniz Institut für Alternsforschung-Fritz Lipmann Institut, 07745 Jena, Germany; zahra.fanaeikahrani@leibniz-fli.de

**Keywords:** Klotho, aging, N-glycosylation, ER-Golgi trafficking, ER chaperone

## Abstract

Klotho is an anti-aging protein whose deletion significantly reduces lifespan in mice, while its over-expression increases lifespan. Klotho is a type-I transmembrane protein that is N-glycosylated at eight positions within its ectodomain. Our study demonstrates that N-glycosylation or mutation at position N614, but not at N161, N285, or N346 in mouse Klotho, is critically involved in the transport of Klotho out of the endoplasmic reticulum (ER). Consequently, while wild-type Klotho-EGFP as well as the N-glycosylation mutants N161Q, N285Q, and N346Q were present at the plasma membrane (PM), only small amounts of the N614Q Klotho-EGFP were present at the PM, with most of the protein accumulating in the ER. Protein interactome analysis of Klotho-EGFP N614Q revealed increased interactions with proteasome-related proteins and proteins involved in ER protein processing, like heat shock proteins and protein disulfide isomerases, indicative of impaired protein folding. Co-immunoprecipitation experiments confirmed the interaction of Klotho-EGFP N614Q with ER chaperons. Interestingly, despite the low amounts of Klotho-EGFP N614Q at the PM, it efficiently induced FGF receptor-mediated ERK activation in the presence of FGF23, highlighting its efficacy in triggering downstream signaling, even in limited quantities at the PM.

## 1. Introduction

Found serendipitously in 1997 [1], Klotho is now one of the most studied age-related proteins, with PubMed listing more than 3700 publications in early 2024. Klotho is a membrane-bound protein that serves as a co-receptor for fibroblast growth factor 23 (FGF23) [2,3]. Klotho is subject to posttranslational modifications such as ectodomain shedding and glycosylation. Ectodomain shedding is mediated by the cleavage of the extracellular domain of Klotho by proteases [4,5], and it gives rise to the release of “shed Klotho” (sKlotho) from the plasma membrane (PM) [6]. N-glycosylation is an important process in cells critical for endoplasmic reticulum (ER)-to-Golgi trafficking, proper protein folding, quality control, cell–cell adhesion of glycoproteins, and protection against extracellular proteinases [7]. Previous studies have demonstrated that Klotho undergoes both N-glycosylation and O-glycosylation [6,8]. Human Klotho possesses eight N-glycosylation sites, N73, N126, N250, N311, N574, N579, N597, and N661 [8], which are conserved in mice (N108, N161, N285, N346, N609, N614, N632 and N696). Zhong and colleagues discovered that when expressing a secreted, truncated Klotho in Chinese hamster ovary (CHO) cells or human embryonic kidney (HEK) 293 cells, these two different cell types produced sKlotho with different forms of glycosylation [8]. Notably, highly sialylated N-glycans were detected in CHO-sKlotho, while HEK-sKlotho was enriched with disaccharide N, N-di-N-acetyllactose diamine (LacdiNAc) N-glycan at multiple N-linked sites. Surprisingly, CHO-expressed Klotho was less active as a fibroblast growth factor receptor (FGFR) co-receptor despite showing improved pharmacokinetic properties. Furthermore, N-glycosylation in Klotho is crucial for its proper folding and secretion [8]. However, it remains unclear how these particular glycosylation sites are involved in Klotho’s folding and its function as a co-receptor.

There has been great interest in developing Klotho-based therapeutic tools and biomarkers; therefore, a proper understanding of the function of glycosylation sites is crucial for the development of efficacious therapeutic Klotho variants [9]. In the current study, we examined the role of N-glycosylation in the membrane transport and cellular function of murine Klotho. To find out the role of specific glycosylation sites, we changed asparagine (N) to glutamine (Q) at four different sites (N161Q, N285Q, N346Q, and N614Q). These sites were chosen because their human counterparts showed normal secretion (N161 and N346) or secretion deficits (N285 and N614) in a soluble Klotho construct [8]. Our data suggest that the N-glycosylation of Klotho is important for its folding, membrane transport, and protein–protein interactions, which ultimately lead to the activation of specific signaling pathways. Specifically, murine Klotho N614Q (corresponding to N579 in human Klotho) is transported less to the PM, resulting in decreased cell surface expression and secretion. Interactome analysis revealed a significant increase in interactions between Klotho N614Q and components of the proteasome machinery, suggesting a potential mechanism that removes misfolded Klotho N614Q to maintain cell homeostasis. Moreover, Klotho N614Q specifically interacted with proteins involved in ER protein processing, especially chaperones, suggesting that N-glycosylation at N614 is essential for proper protein folding.

## 2. Materials and Methods

### 2.1. Chemical Compounds

Dulbecco’s modified Eagle Medium + GlutaMax (Gibco, Waltham, MS, USA, 61965-026), Hygromycin B (50 mg/mL) (Thermo Fischer Scientific, 10687010, Waltham, MS, USA), penicillin/streptomycin (Merck, P0781, Rahway, NJ, USA), Fetal Bovine Serum (F7524 Sigma-Aldrich, Burlington, MA), ROTI^®^Histofix 4% paraformaldehyde (Carl Roth, No. P087.1, Karlsruhe, Germany), Hoechst 33342 (Invitrogen, H1399, Waltham, MS, USA), Human FGF23 protein (R&D Systems, 2604, Minneapolis, MN, USA), Lipofectamine™ 2000 Transfection Reagent (Thermo Fischer Scientific, 11668019), Protease Inhibitor Cocktail (Sigma P8340, Burlington, MA, USA), phosphatase inhibitors (PhosSTOP, Roche, #4906837001, Basel, Switzerland), CHAPSO (Carl Roth), I-Block™ powder (T2015, Applied Biosystems, Thermo Fisher Scientific). For a list of antibodies used see Table 1.

### 2.2. Construction of Klotho Mutants

Four glycosylation sites in mouse Klotho, including N161, N285, N346, and N614, were chosen for mutation to glutamine (Q). For this purpose, a mouse Klotho C-terminally tagged with EGFP in pcDNA3.1 Hygro was used. The Quick-change Site-Directed Mutagenesis Kit (Invitrogen) was employed using the primers listed in Table 2.

### 2.3. Generation of Cells Stably Expressing Klotho-EGFP Variants

HEK293T (RRID: CVCL_0063) was grown in Dulbecco’s modified Eagle Medium + GlutaMax, supplemented with 10% FBS and 1% penicillin/streptomycin, and incubated at 37 °C in a 5% CO_2_ atmosphere with 95% relative humidity.

Plasmids encoding Klotho-EGFP, or its glycosylation variants were transiently transfected into HEK293T cells using Lipofectamine 2000 reagent, as specified by the manufacturer. Stable cell lines were established using Hygromycin B (0.1 mg/mL). After 16 to 17 days of selection, resistant colonies were pooled and further cultured. Next, cells highly expressing GFP were sorted using fluorescence-activated cell sorting (FACS) and subcultured.

### 2.4. Choroid Plexus Isolation and Lysis

Wild-type mice were killed in a CO_2_ chamber (0.5 L/min). To isolate the choroid plexus (CP) from the fourth ventricle, the brain was first isolated, and the cerebellum was separated from the cerebrum using a scalpel. The cerebellum was then detached from the brainstem, revealing the fourth ventricle cavity and exposing the internal choroid plexus, which was carefully extracted. For the isolation of the CP from the lateral ventricles, the cerebral hemispheres were separated with a blade, and the ventricles were opened under a microscope. The transparent CP, located opposite the olfactory region, was then gently removed. CP tissue was homogenized using the Precellys Keramik Kit 91-PCS-CK14 (Peqlab Biotechnology, BERT03961-1-103, Erlangen, Germany), along with ten Precellys ceramic beads and 80 μL of STEN lysis buffer containing 1:500 Protease Inhibitor Cocktail (PIC, Sigma P8340), and transferred into a 2 mL Precellys tube. The tubes were then placed in the Bertin Precellys 24 Tissue Homogenizer (Peqlab Biotechnology, Erlangen, Germany) and homogenized in two 30-s steps at 5100 rpm, followed by incubation on ice for 30 min. The homogenates were subsequently centrifuged at 4 °C for 5 min at 13,200 rpm, and the supernatant was used for further experiments.

### 2.5. Cell Lysis and Western Blotting

For Western Blotting, the medium from HEK293T cells expressing Klotho-EGFP variants was collected, and the cells were lysed using STEN lysis buffer (50 mM Tris-HCl pH 7.6, 150 mM NaCl, 2 mM EDTA, 0.2% NP-40) supplemented with protease and phosphatase inhibitors. Cell lysates were separated on an 8% SDS-PAGE gel, and the secreted form of Klotho was detected by loading the culture media onto a 10% SDS-PAGE gel. Proteins were transferred to a PVDF membrane, blocked with 5% BSA in TBS buffer containing 0.1% Tween-20 or I-block (1 g of I-Block^™^ powder (T2015, Applied Biosystems, Thermo Fisher), 0.1% Tween-20 (9127.2, Carl Roth) in 500 mL of 1 × PBS) and incubated overnight at 4 °C with antibodies as listed in Table 1. After three 10-min washes with 1× TBS-Tween, the membranes were incubated with the secondary antibody for 60 min at room temperature. Following incubation, the membranes were washed again (3 × 10 min with 1× TBS-Tween). Immunoreactive protein bands were then detected using enhanced chemiluminescence (ECL) solution. To visualize protein bands, chemiluminescent imaging was performed using the Cytiva Amersham ImageQuant 800 Western blot imaging system.

### 2.6. Deglycosylation Assay

Endoglycosidase H (EndoH) treatment was performed according to the manufacturer’s instructions (New England Biolabs, P0702L, Ipswich, MA, USA). Briefly, 30 μg of cell lysate was combined with 1 μL of 10× glycoprotein denaturing buffer and H_2_O to prepare a total reaction volume of 10 μL and denatured by heating at 100 °C for 10 min. Next, 2 μL of 10 × GlycoBuffer 3, 2 μL Endo H, and H_2_O were added to the mixture to bring the total reaction volume to 20 μL. The mixture was incubated at 37 °C for 1 h before being processed for SDS-PAGE and Western blotting. For PNGase F treatment, the protocol for denaturing conditions according to the manufacturer’s instructions was followed. Briefly, 30 μg of cell lysate was combined with 1 μL of 10 × glycoprotein denaturing buffer and H_2_O to prepare a total reaction volume of 10 μL and denatured by heating at 100 °C for 10 min, followed by rapid cooling on ice. Next, the reaction was adjusted to a total volume of 20 µL by adding 2 µL of 10 × GlycoBuffer 2, 2 µL of 10% NP-40, 6 µL of H_2_O, and 1 µL of PNGase F (New England Biolabs, P0704L). Then, it was incubated at 37 °C for 1 h and processed for SDS-PAGE and Western blotting.

### 2.7. Immunofluorescence and Cell Surface Staining

For immunofluorescence staining, cells were cultured on coverslips pre-treated with poly-L-lysine for 1 h. Afterward, cells on coverslips were fixed with ROTI^®^Histofix 4% paraformaldehyde at room temperature for 20 min. After washing with PBS, the fixed cells were treated with 50 mM NH4Cl (10 min) and later with 0.2% Triton X-100 for 2 min and then incubated with a blocking medium (1 mL of FCS, 1 g of BSA, and 0.1 g of fish gelatin in 100 mL of 1 × PBS) for 10 min. Subsequently, the coverslips were incubated with the respective primary antibody (see Table 2) for 30 min. After two washes with PBS, the coverslips were incubated with the secondary antibodies for 20 min. Next, the nuclei were stained using Hoechst 33342. The coverslips were mounted, and images were obtained using a 63 × objective on an Axiovert 200 ApoTome microscope and ZEN3.7 software (Carl Zeiss Microscopy GmbH, Jena, Germany).

For surface staining, cells grown on coverslips were washed three times with 1 × PBSI (PBS containing 1 mM CaCl_2_ and 0.5 mM MgCl_2_). The coverslips were then incubated with anti-Klotho antibody (AF1819, R&D Systems, 1:50 dilution) for 30 min at 4 °C, rinsed 3 times with PBSI, and fixed with ROTI^®^Histofix 4% paraformaldehyde on ice for 5 min, followed by room temperature fixation for 20 min. Cells were blocked with 1% BSA in 1 × PBS for 20 min at room temperature and probed with donkey anti-mouse Alexa fluor 555 secondary antibody, followed by 3 washes with PBSI. Nuclear staining and imaging were performed as described above.

### 2.8. Surface Biotinylation

A total of 3 × 10^6^ cells from each cell line were seeded onto four 10 cm plates and cultured for two days. The plates were placed on ice and washed three times with 5 mL of ice-cold PBSI. They were then incubated with 3 mL of 0.5 mg/mL EZ-Link™-Sulfo-NHS-LC-biotin (Thermo Fischer, 21335) for 30 min, followed by four washes with 5 mL of ice-cold 20 mM glycine in PBS to quench the biotinylation reaction. The plates were then rinsed with ice-cold PBS containing 1 mM Ca^2+^ and 0.5 mM Mg^2+^ before being lysed in 50 mM Tris-HCl pH 7.6, 150 mM NaCl, 2 mM EDTA, 1% NP-40, and a protease inhibitor cocktail. Lysates were incubated on ice for 30 min, followed by centrifugation at 6000× *g* at 4 °C for 6 min. For each sample, 90 µL of streptavidin-agarose beads (Merck, GE17-5113-01) were washed twice in lysis buffer for equilibration. Next, 700–800 µg of lysates were added to the equilibrated streptavidin-agarose beads, followed by incubation at 4 °C with rotation (15 rpm). The following day, the beads were washed once with 1 mL of 50 mM Tris-HCl pH 7.6, 325 mM NaCl, 2 mM EDTA, 0.2% NP-40, five times with 1 mL of 50 mM Tris-HCl pH 7.6, 150 mM NaCl, 2 mM EDTA, 1% NP-40, and 1% SDS, and five times with 1 mL of 50 mM Tris-HCl pH 7.6, 150 mM NaCl, and 2 mM EDTA. The beads were resuspended in 2 × SDS sample buffer containing 3 mM biotin and analyzed via Western blotting. Four biological replicates were performed.

### 2.9. Characterization of FGFR Signaling Pathway

For the analysis of the activation of the FGFR signaling pathway, HEK293T cells were seeded (10^6^ cells/well) into a 6-well culture plate until they reached 70–80% confluency. Next, the cells were starved in a serum-free medium for 16 h, followed by the addition of FGF23, and incubated at 37 °C as indicated. After incubation, the cells were kept on ice, washed with cold PBS to inhibit further reactions, and analyzed via Western blotting. Four biological replicates were performed.

### 2.10. Immunoprecipitation

For GFP-trap immunoprecipitations, cells grown in 10 cm plates were rinsed with cold PBS and then lysed using CHAPSO buffer (150 mM citrate buffer, pH 6.4, 2% CHAPSO) supplemented with protease and phosphatase inhibitors. GFP-TRAP agarose beads (Chromotek, Martinsried, Germany) were equilibrated by washing with 150 mM citrate buffer three times for 5 min at 4 °C. A total of 1.2 mg of lysates were incubated with the GFP-Trap overnight at 4 °C while slowly rotating. The supernatants were then collected, and the beads were washed twice with 500 µL of wash buffer (150 mM citrate buffer, pH 6.4, 0.5% CHAPSO) and once with 500 µL of 150 mM citrate buffer and collected by spinning down at 2500× *g* for 5 min at 4 °C. Proteins were eluted from the beads by boiling in 2 × Laemmli buffer (SDS sample buffer) at 95 °C for 10 min. Finally, the samples were resolved using SDS-polyacrylamide gel electrophoresis on 8% gels. Three biological replicates were performed.

### 2.11. LC–MS Proteomic Analysis of Klotho Interactors

To investigate the proteomic interactors of Klotho-EGFP variants, we conducted immunoprecipitation assays on untreated cell samples, as described in the previous section. The elution buffer (200 mM glycine, pH 2.5) was added to the beads containing the immunoprecipitated protein complexes. Next, the mixture was pipetted up and down for approximately 30 s to ensure thorough mixing. The bead–buffer mixture was then transferred to Pierce Spin Columns with Snap Caps. The columns were placed in new low protein-binding tubes (Eppendorf^®^ Protein LoBind tubes, 1.5 mL, EP0030108116-100EA, Hamburg, Germany), and the samples were centrifuged at 300× *g* for a few seconds to collect the eluate. To neutralize the eluate, 5 µL of 1 M Tris (pH 10.4) was added to each sample. To prepare the samples, 52.5 µL of 3 × lysis buffer containing 4% SDS, 100 mM HEPES (pH 8.5), and 50 mM DTT were added to each sample. The mixture was then boiled at 95 °C for 7 min and sonicated. Next, reduction and alkylation processes were carried out by treating the samples with 15 mM iodoacetamide for 30 min at room temperature in the dark. Afterward, phosphoric acid (2.5% final concentration) was used to acidify the samples; then, they were diluted with seven volumes of S-trap binding buffer (100 mM TEAB, 90% methanol). Proteins were bound to a 96-well S-trap microplate (Protifi, Fairport, NY, USA), washed three times with binding buffer, and digested with trypsin (1 µg/sample) in 50 mM TEAB, pH 8.5, for 1 h at 47 °C. Elution was performed in three steps using 50 mM TEAB, 0.2% formic acid, and 50% acetonitrile with 0.2% formic acid. The eluates were dried with a speed vacuum centrifuge (Eppendorf Concentrator Plus) and stored at −20 °C. Samples were loaded onto Evotips (Evosep) according to the manufacturer’s instructions, involving sequential washing, conditioning, equilibration, and sample loading steps. Peptides were separated using an Evosep One system with a 15 cm × 150 μm i.d. column packed with 1.9 μm Reprosil-Pur C18 beads. The 44-min gradient (30 samples per day) used water with 0.1% formic acid and acetonitrile with 0.1% formic acid as solvents. The LC was coupled to an Orbitrap Exploris 480 mass spectrometer via a PepSep Emitter heated to 300 °C with a 2 kV spray voltage. DIA data acquisition included full MS scans (350–1650 *m*/*z*) at a resolution of 120,000 FWHM, with DIA scans using 40 variable-width mass windows and HCD fragmentation (normalized collision energy of 29%). MS/MS spectra were acquired at 30,000 FWHM with a fixed first mass of 200 *m*/*z*, 1 × 10^6^ ion accumulation or a 45 ms fill time limit. Data were acquired in profile mode using Xcalibur 4.5 and Tune 4.0 software. All proteomics experiments were conducted using six biological replicates for each cell line.

### 2.12. Proteomic Data Processing

DIA raw data were processed with Spectronaut (v.18, Biognosysis) using the directDIA pipeline and searched against a Mus musculus SwissProt database (16,748 entries) with added contaminants (247 entries). Identifications were filtered to achieve a 1% FDR at both peptide and protein levels. Relative quantification in Spectronaut was performed for each paired comparison using the replicates from each condition. The data were then exported from Spectronaut, and further data analyses and visualization were conducted with R studio, the Enrichr server (https://maayanlab.cloud/Enrichr/) (access date on 8 February 2024), and Microsoft Excel (version 2403). A cut-off of *p*-value ≤ 0.05 and absolute log2 fold chain ≥ 0.58 in was applied to select the most significantly enriched proteins.

### 2.13. Whole-Cell Proteomics and Phosphoproteomics

The initial steps for whole-cell proteomics and phosphoproteomics followed the same protocol used for LC–MS identification of protein interactors. After alkylation with iodoacetamide, the lysates were precipitated overnight at −20 °C by adding four volumes of ice-cold acetone. The next day, the samples were centrifuged at 20,800× *g* for 30 min at 4 °C. The pellets were washed twice with ice-cold 80% (*v*/*v*) acetone in water, followed by centrifugation at 20,800× *g* at 4 °C for 10 min. After removing the acetone, the pellets were air-dried and resuspended in 100 µL of digestion buffer (1 M guanidine chloride in 100 mM HEPES, pH 8). The samples were sonicated as previously described, then LysC (Wako) was added at an enzyme-to-protein ratio of 1:100 (*w*/*w*), and digestion was carried out for 2 h at 37 °C with shaking at 650 rpm. The samples were then diluted 1:1 with Milli-Q water, and trypsin (Promega) was added at an enzyme-to-protein ratio of 1:100 (*w*/*w*). The digestion continued overnight at 37 °C with shaking at 650 rpm. The following day, the digests were acidified by adding TFA to a final concentration of 1% (*v*/*v*) and desalted using a Waters Oasis^®^ HLB µElution Plate (30 µm; Waters Corporation, MA, USA) under soft vacuum, following the manufacturer’s instructions.

Phosphorylated peptides were enriched using Fe(III)-IMAC cartridges (Agilent) in an automated fashion, following the standard protocol of the AssayMAP Bravo Platform (Agilent Technologies, Santa Clara, CA, USA). For LC–MS of phosphorylated proteins, samples were dissolved in MS Buffer containing 5% acetonitrile, 95% Milli-Q water, and 0.1% formic acid and were spiked with iRT peptides (Biognosys, Zürich, Switzerland). Peptides were separated in trap/elute mode using the nanoAcquity MClass Ultra-High Performance Liquid Chromatography system (Waters Corporation, Milford, MA, USA), which was equipped with a trapping column (nanoEase M/Z Symmetry C18, 5 μm, 180 μm × 20 mm) and an analytical column (nanoEase M/Z Peptide C18, 1.7 μm, 75 μm × 250 mm). Solvent A was composed of water with 0.1% formic acid, while solvent B consisted of acetonitrile with 0.1% formic acid. An initial 5 μL portion of the sample was introduced onto the trapping column at a consistent flow rate of 5 μL/min using solvent A. Subsequently, peptides were separated through the analytical column at a steady flow rate of 0.3 μL/min. Throughout the elution process, the concentration of solvent B was gradually increased from 0% to 40% in a non-linear gradient over 60 min, resulting in a total duration of 75 min for the entire run. All phosphoproteomics experiments were conducted using six biological replicates for each cell line and each condition, including samples treated with or without FGF23.

### 2.14. Data Analysis for Phosphoproteomics

To conduct a thorough analysis of phosphoproteomics data, various modifications were considered. These included Carbamidomethyl (C) as a fixed modification and Oxidation (M), Acetyl (Protein N-term), and Phospho (STY) as variable modifications. The localization probability for post-translational modifications (PTMs) was set at 0.75, with phosphosite consolidation being sum-based. Up to 2 missed cleavages for trypsin and a maximum of 5 variable modifications were permitted in the identification process. Relative quantification at the phosphosite level was carried out using Spectronaut, comparing replicate samples from each experimental condition. Subsequently, the data, including tables of candidates and protein quantities, were exported for further analysis and visualization using RStudio, along with customized pipelines and scripts. Significant phosphosites were chosen based on a log2 fold change (log2FC) cutoff of 0.58 and a q-value < 0.05. An adapted version of PhosR [10] was employed in the analysis to pinpoint the most regulated kinases and their top phosphosites.

### 2.15. Statistics

In this study, the quantification of protein band expression levels was performed utilizing ImageJ software (ImageJ 1.54f). We conducted statistical analysis utilizing Microsoft Excel (version 2403) to assess the variance in protein expression observed on Western blots. An independent samples *t*-test was employed, with the assumption of equal variances across groups. A significance threshold of *p* ≤ 0.05 was utilized to determine statistical significance. Also, the standard deviation (STDEV) was calculated using the same software.

## 3. Results

### 3.1. Klotho Is an N-Glycosylated Transmembrane Protein

To confirm the N-glycosylation of Klotho, lysate from mouse choroid plexus (which expresses high levels of Klotho) was prepared and treated with Peptide-*N*-Glycosidase F (PNGaseF) or Endoglycosidase H (endoH) and analyzed through Western blot using an anti-Klotho antibody (Figure 1A). PNGaseF eliminates N-linked oligosaccharides from glycoproteins, whereas endoH removes high-mannose but not complex-glycosylated sugar moieties. EndoH sensitivity characterizes the ER-resident forms of a protein, while EndoH resistance indicates localization at or beyond the Golgi apparatus. As shown previously, Klotho is present as a double band on Western blot [5,6]. The lower band corresponds to the immature, ER-resident form because it is EndoH-sensitive, as indicated by its shift to the position of the deglycosylated band (Figure 1A, deg). The upper band is EndoH-resistant, corresponding to the mature form (mat) that localizes to Golgi and/or PM. As a tool to analyze N-glycosylation mutants, we used a C-terminally enhanced green fluorescent protein (EGFP)-tagged mouse Klotho and generated stably expressing human embryonic kidney (HEK)293T cells. Lysates expressing Klotho-EGFP were subjected to EndoH and PNGaseF digest and subsequent Western blot (Figure 1B). Like endogenous untagged Klotho, stably expressed Klotho-EGFP is present in an immature, EndoH-sensitive, ER-resident form (imm) and a mature (mat), EndoH-resistant Golgi/PM-localized form. The ratio of imm/mat is similar to that of the endogenous protein, demonstrating comparable localization.

### 3.2. N614 Is Essential for Surface Transport and Secretion of Klotho

To identify the role of the depicted glycosylation sites (Figure 1C,D), we generated HEK293T cells stably expressing the Klotho-EGFP glycosylation mutants N161Q, N285Q, N346Q, and N614Q, respectively. First, the intracellular localizations of the Klotho-EGFP variants were assayed through cell surface immunofluorescence using an antibody against the Klotho ectodomain (Figure 2A). Klotho-EGFP was present in intracellular compartments resembling the ER and was transported to the PM, as indicated by its surface staining. Klotho-EGFP N161Q, N285Q, and N346Q displayed similar localization patterns, suggesting these N-glycosylation sites are not essential for surface transport. In contrast, Klotho-EGFP N614Q showed only an ER-like distribution and no cell surface staining. Higher exposure times and the comparison with untransfected HEK293 cells revealed very low amounts of Klotho-EGFP N614Q on the surface, suggesting that transport to the PM is strongly, but not fully, inhibited (Figure 2B). No endogenous Klotho was detected under these conditions.

To further support these findings, we analyzed the glycosylation and secretion of Klotho-EGFP variants using Western blotting (Figure 2C). Intracellular Klotho-EGFP WT, as well as glycosylation mutants N161Q, N285Q, and N346Q, showed the typical appearance of a double band corresponding to mature and immature Klotho. The mature Klotho-EGFP corresponded to 40–45% of the total protein. In contrast, in cells expressing Klotho-EGFP N614Q, the mature band represented only 15% of the total protein, indicating a transport defect (Figure 2D). Furthermore, analysis of the cell medium showed that cells expressing Klotho-EGFP N614Q did not secrete any sKlotho, in contrast to cells expressing Klotho-EGFP WT and all the other glycosylation mutants, which secreted sKlotho at similar levels. To quantify the respective amounts of Klotho-EGFP WT vs. N614Q at the PM, we next performed cell surface biotinylation using a non-permeable biotin reagent (Figure 2E,F). While robust amounts of Klotho-EGFP WT were detected at the PM, 4-fold less was detected in the case of Klotho-EGFP N614Q (Figure 2F). EEA1 served as a cytosolic protein control that is not biotinylated in the surface fraction, indicating specific surface-biotinylation.

To confirm that Klotho-EGFP N614Q is not transported out of the ER, lysates of cells stably expressing Klotho-EGFP variants were subjected to PNGaseF or EndoH digestion, followed by SDS-PAGE and Western blotting (Figure 3). In the case of Klotho-EGFP WT and the glycosylation mutants N161Q, N285Q, and N346Q, the slower migrating band was EndoH-resistant, indicating transport to the Golgi and beyond. The lower, immature band was EndoH-sensitive and shifted to the position of the deglycosylated Klotho-EGFP, indicating ER residence. In contrast, the single band of Klotho-EGFP N614Q was completely EndoH-sensitive, confirming that this variant is mostly retained in the ER (Figure 3A). Further support for predominant ER localization came from co-immunostaining of Klotho-EGFP WT and the N614Q mutant with the ER marker protein disulfide isomerase (PDI, Figure 3B). While Klotho-EGFP is present at the PM and the ER, Klotho-EGFP N614Q mainly colocalizes with PDI, indicating predominant ER localization. Taken together, these data indicate that glycosylation at N614 is essential for the transport of Klotho to the PM and its subsequent secretion.

### 3.3. Glycosylation at N614 Is Essential for the Folding of Klotho

To get more insights into the consequences of inhibiting N-glycosylation at N614, we determined the interactomes of both Klotho-EGFP WT and N614Q variants. We used GFP-TRAP immunoprecipitation (IP) followed by mass spectroscopy, as schematized in Figure 4A. A total of 976 proteins significantly interacted with Klotho-EGFP (Appendix A), whereas 1874 proteins significantly interacted with Klotho-EGFP N614Q (Appendix A). Of these, 1295 proteins differentially interacted with Klotho-EGFP N614Q compared to Klotho-EGFP WT (Appendix A). Notably, two proteins significantly interacted with Klotho-EGFP but not with Klotho-EGFP N614Q (Appendix A).

KEGG pathway analysis of the 1295 proteins interacting with Klotho-EGFP N614Q indicated that, compared to Klotho-EGFP WT, Klotho-EGFP N614Q preferentially interacted with proteasomal proteins and proteins involved in ER processing (Figure 4B,C, Appendix A). These findings are indicative of a compromised protein folding process for the N614Q variant in the ER, followed by ER-associated degradation via the proteasome. Remarkably, among the top 10 ER-associated proteins that preferentially interacted with Klotho-EGFP N614Q, seven (HSPA4, PDIA4, PDIA3, HSPA5, P4HB, CALR, and MOGS) are chaperones [11,12], underscoring the N614Q variant’s aberrant folding and its increased engagement with the ER chaperone machinery (Figure 4C, Appendix A). Among these chaperones, MOGS (mannosyl-oligosaccharide glucosidase) stands out as a crucial player involved in the quality control of glycoproteins [13]. We next validated the increased interaction of Klotho-EGFP N614Q with two ER-related proteins, HSPA5 (BiP) and ERGIC53, through co-IP and Western blot analysis (Figure 4D,E). Klotho-EGFP N614Q interacted much more strongly with HSPA5 and ERGIC53 than Klotho-EGFP WT, confirming the MS data.

### 3.4. Impaired Surface Transport of Klotho-EGFP N614Q Does Not Block Activation of the FGFR Signaling Pathway

Klotho interacts with FGFR and FGF23, and this interaction activates the FGFR signaling pathway [2,3]. Therefore, we determined the effect of our Klotho deglycosylation mutants on FGFR signaling. To this end, HEK293T cells, with or without stably expressing Klotho-EGFP variants, were treated with EGF or FGF23, and the phosphorylation state of ERK (Extracellular signal-regulated kinase) was determined via Western blot. As shown previously [2], in HEK293 cells that do not express Klotho, ERK1/2 was activated by EGF but not by FGF23, as indicated by elevated levels of phosphorylated ERK1/2 (Figure 5A). In contrast, in HEK293T cells overexpressing Klotho-EGFP, the ERK1/2 pathway was activated by FGF23, likely via Klotho/FGFR signaling. Similarly, Klotho-EGFP glycosylation mutants N161Q, N285Q, and N346Q also activated ERK1/2 signaling (Figure 4A, B). Interestingly, in cells expressing Klotho-EGFP N614Q, which is mainly trapped in the ER (Figure 2 and Figure 3), FGF23 activated ERK1/2 signaling to the same extent as in cells expressing Klotho-EGFP WT (Figure 5A,B). These data suggest that N-glycosylation at positions 161, 285, 346, or 614 is not essential for the interaction of Klotho with FGFR in the presence of FGF23. The minute amounts of Klotho-EGFP N614Q at the PM may be sufficient for the initiation and activation of FGFR signaling.

### 3.5. ERK Activation by Minute Amounts of Klotho-EGFP at the Plasma Membrane

Considering the efficient activation of FGFR by the mostly ER-retained Klotho-EGFP N614Q, we evaluated whether this could result from internalized FGF23 that reached the ER during the 5-min incubation. To this end, we reduced the incubation time with FGF23. As shown in Figure 6C, an incubation with FGF23 for 1 min was sufficient to activate ERK signaling in all conditions. The 1-min incubation ruled out the uptake of FGF23 and transport to the ER, suggesting that the small amount of N614Q at the PM, not the ER-retained major fraction, is responsible for FGFR activation. If very small amounts of Klotho-EGFP at the PM are sufficient to trigger robust ERK activation, one would expect that lowering the dose of FGF23 would not ameliorate ERK activation. This seems to be the case since reducing FGF23 concentration to 8% of the original dose still elicited strong ERK activation, suggesting that the system is already saturated at very low levels of Klotho (and possibly FGFR) at the PM (Figure 6A,B).

Alternatively, Klotho-EGFP N614Q may possess a higher binding affinity to FGFR, potentially resulting in increased pathway activation even with minimal protein amounts at the PM. To investigate this, GFP-TRAP co-IP was employed, precipitating Klotho-EGFP followed by Western blotting with FGFR3 and Klotho antibodies (Figure 6D). Quantification of FGFR3 bound to Klotho-EGFP revealed a weaker interaction between FGFR3 and Klotho-EGFP N614Q compared to Klotho-EGFP WT (Figure 6E). These results suggest that ERK activation by the low amounts of surface-localized Klotho-EGFP N614Q cannot be explained by increased FGFR3 binding. However, it is interesting to note that FGF23 appears to increase the binding of Klotho-EGFP N614Q to FGFR3 more than it does with Klotho-EGFP (Figure 6E).

Next, we tested if very low amounts of Klotho-EGFP at the PM could elicit robust FGF23-induced ERK activation. To this end, we FACS-sorted HEK293T cells stably expressing Klotho-EGFP into three subpools expressing low (l), median (m), and high (h) levels of Klotho-EGFP and subjected them to FGF23-induced ERK- activation (Figure 7). In all Klotho-EGFP(l) cells, ERK could be strongly stimulated by FGF23 (Figure 7). This demonstrates that very limited amounts of Klotho-EGPF at the PM, as observed in Klotho-EGFP N614Q-expressing cells, can robustly activate ERK.

### 3.6. Phosphoproteomics Analysis Confirms Activation of the MAPK Signaling Pathway by Klotho-EGFP WT and N614Q Variant

To gain deeper insights into the downstream pathways activated by FGF23, we conducted a phosphoproteomics analysis of HEK293 cells stably expressing Klotho-EGFP WT or the N614Q variant, with or without treatment with FGF23. HEK293 cells expressing only EGFP were used as a control. Phosphorylated proteins and their phosphosites were compared in each cell line, and phosphoproteins that were differentially regulated in EGFP control cells were excluded from further analysis. Following FGF23 treatment, 300 phosphosites of 248 phosphorylated proteins were upregulated in Klotho-EGFP expressing cells (Appendix A). In the Klotho-EGFP N614Q mutant, 343 phosphosites of 276 phosphorylated proteins were differentially up-regulated after FGF23 activation (Appendix A). Of these, 137 proteins were similarly phosphorylated after FGF23 induction in both Klotho-EGFP WT and N614Q expressing cell lines (Figure 8A, Appendix A). Differentially phosphorylated proteins were subjected to Gene Ontology (GO) and pathway enrichment analyses utilizing BioPlanet pathways [14] from the Enrichr website (https://maayanlab.cloud/Enrichr/), accessed 8 February 2024. In both Klotho-EGFP WT and N614Q expressing cells, the mitogen-activated protein kinase (MAPK) signaling pathway was activated after FGF23 treatment (Figure 8B, Appendix A). Specifically, ELK1, a direct downstream target of ERK [15], was phosphorylated after FGF23 treatment in both Klotho-EGFP WT and N614Q-expressing cells. The Y187 phosphosite in MAPK1 (ERK1) was phosphorylated in cells expressing Klotho-EGFP N614Q but not in those expressing WT. Both findings confirmed the activation of the ERK signaling pathway by FGF23 in the presence of Klotho. Additionally, several proteins associated with MAPK signaling, including SOS1, RPS6KA3, FLNA, DUSP16, and HSPB1, were phosphorylated following FGF23 stimulation in both Klotho-EGFP WT and N614Q variants (Appendix A).

Next, phosphosites that were up-regulated in each cell line were used to predict kinase–phosphosite relationships utilizing the R package PhosR (https://pyanglab.github.io/PhosR/ access on 8 February 2024). PhosR evaluates each phosphosite against a range of kinases, predicting the most likely kinase responsible for the phosphorylation of that specific phosphosite (Appendix A). Consequently, the top three phosphosites for each kinase are shown in a heatmap (Figure 8C,D). The prediction of upstream kinases in both Klotho-EGFP WT and N614Q cells suggested the involvement of various MAPK proteins in phosphorylating the identified phosphosites. This suggests that not only Klotho-EGPF but also the N614Q variant is activated by the canonical FGF23-FGFR-Klotho signaling pathway starting from the cell surface.

## 4. Discussion

To determine the role of glycosylation in Klotho membrane transport and function, we individually disrupted four glycosylation sites by replacing asparagine with glutamine in Klotho-EGFP. Analysis of HEK293T cells stably expressing the variants demonstrated that N-glycosylation at N614, but not at N161, N285, or N346, is critical for the folding of Klotho-EGFP and its transport from the ER to the PM. Consequently, Klotho-EGFP N614Q, which colocalized with an ER marker, was EndoH sensitive, and no detectable amounts of Klotho-EGFP N614Q were secreted. Preventing N-glycosylation at N614 may inhibit the transport of properly folded Klotho-EGFP, for example, by affecting binding to the lectin-type ER-Golgi transporter ERGIC53 [16]. However, we found that glycosylation at N614 does not reduce binding to ERGIC53. Alternatively, preventing N-glycosylation at N614 or changing asparagine to glutamine could induce misfolding or prevent the proper folding of Klotho-EGFP. Because of the strict quality control in the ER [17], a misfolded Klotho-EGFP will not leave the ER. Our MS-based data shows an increased interaction of Klotho-EGFP N614Q with several ER chaperones like HSPA4, PDIA4, PDIA3, HSPA5, P4HB, CALR, and MOGS strongly suggest misfolding of Klotho-EGFP N614Q. This is further supported by the interaction of Klotho-EGFP N614Q with proteasomal subunits, probably caused by misfolding and subsequent ERAD (Endoplasmic Reticulum-Associated Degradation) [18]. N-glycosylation of Klotho has been analyzed using a soluble Klotho construct [8]. In that study, mutating the N-glycosylation site N579 in human Klotho (corresponding to the mouse N614) resulted in a reduced secretion but not an almost full ER retention, as observed in our case. This difference could be explained by the use of different assays (soluble Klotho vs. full-length EGFP-tagged Klotho, transient vs. stable transfection, and human Klotho vs. mouse Klotho).

Our proteomics analysis revealed that, in addition to ER-chaperones, several ribosome-associated proteins interacted more with Klotho-EGFP N614Q compared to Klotho-EGFP WT. Ribosomal proteins may interact with Klotho-EGFP N614Q as part of a quality control mechanism aimed at facilitating its degradation via the proteasome. Such interactions are characteristic of the ribosome-associated quality control (RQC) pathway, which plays a key role in recognizing and processing misfolded or aberrant proteins to prevent their accumulation [19].

To test the significance of Klotho N-glycosylation in FGFR signaling, ERK activation was measured after treatment with FGF23. Interestingly, the ER-retained Klotho-EGFP N614Q elicited an ERK activation comparable to that of Klotho-EGFP. This could indicate that FGF23 is entering the cell and activating FGFR together with Klotho in the ER, which, to our knowledge, has not been previously described. Indeed, when we shortened the incubation time of FGF23 to 1 min, there was no difference in ERK activation compared to longer incubation times. One minute is not sufficient to concentrate FGF23, endocytose it, and sort it via endosomes and the Golgi apparatus to the ER. Could the small amounts of Klotho-EGFP N614Q present in the PM be responsible for the robust activation of ERK? This seems to be indeed the case since reducing the concentration of FGF23 by more than 20-fold (from 2.5 to 0.12 nM) did not reduce ERK activation, either in Klotho-EGFP N614Q or in WT. Additionally, utilizing HEK293T cells with reduced expression of Klotho-EGFP WT did not result in significant alterations in the activation of the FGFR signaling pathway. These data suggest that very small amounts of Klotho at the PM are sufficient to elicit a robust ERK activation, possibly because the low amounts of endogenous FGFR are rate-limiting. A third explanation could be that Klotho-EGFP N614Q has a higher affinity for FGFR, and therefore, reduced amounts at the PM are sufficient for efficient ERK activation. This was ruled out by co-immunoprecipitation of FGFR3 with both Klotho-EGFP WT and the N614Q variant. Our phosphoproteome analysis confirmed that both Klotho-EGFP WT and the N614Q variant activated canonical MAPK-signaling pathways. Future detailed comparisons may uncover novel insights into alternative signaling pathways.

## 5. Conclusions

In conclusion, we show that N614, probably via N-glycosylation, is critical for the folding of mouse Klotho, allowing its export from the ER to the PM. Small amounts of Klotho-EGFP N614Q at the surface are sufficient to robustly induce FGF23-dependent ERK-signaling. N-glycosylation at N161, N285, N346Q, or N614 is not essential for binding to FGFR. Our findings help elucidate the functional differences that result from glycosylation at different sites of Klotho and contribute to our understanding of the cell and molecular biology of the anti-aging protein Klotho.

## Figures and Tables

**Figure 1 cells-13-01743-f001:**
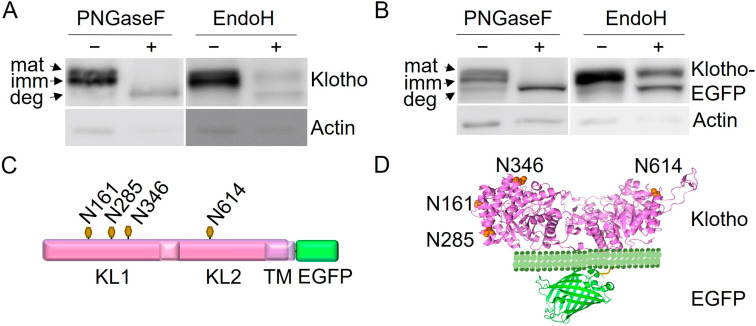
Klotho and Klotho-EGFP are N-glycosylated and present as mature and immature forms. (**A**,**B**) Lysates from mouse Choroid plexus (**A**) or HEK293T cells stably expressing Klotho-EGFP (**B**) were subjected to PNGaseF or EndoH digestion, followed by SDS-PAGE and Western Blotting using antibodies against Klotho and actin. Mat, mature; imm, immature; deg, deglycosylated Klotho or Klotho-EGFP. (**C**) Scheme illustrating the glycosylation sites chosen for mutagenesis within the primary sequence of Klotho-EGFP. (**D**) Three-dimensional model of Klotho-EGFP indicating the precise locations of the mutagenesis sites. (PDB structures: Klotho: 5W21, EGFP: 6YLQ). For full-size blots, see Appendix A.

**Figure 2 cells-13-01743-f002:**
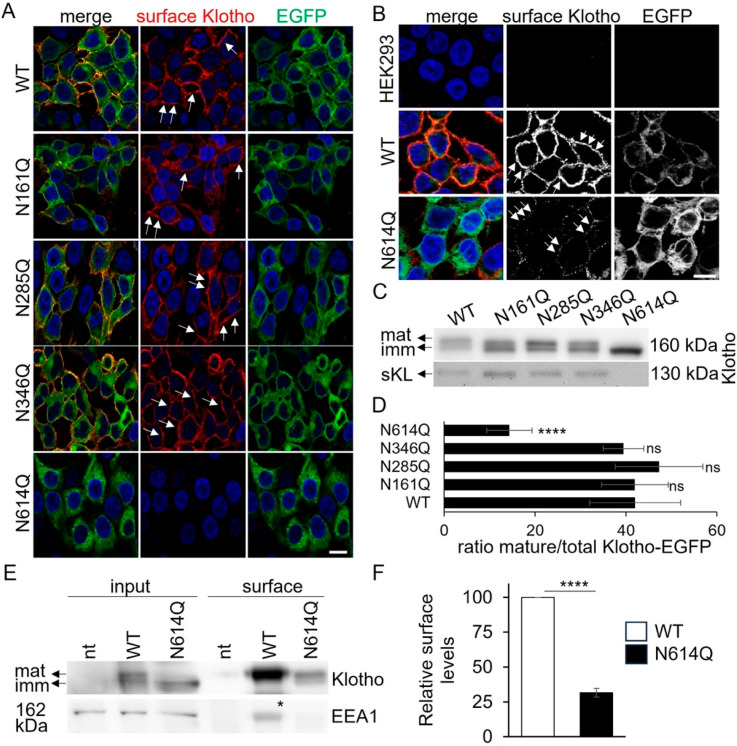
Klotho-EGFP N614Q is not transported to the cell surface. (**A**,**B**) HEK293 cells stably expressing Klotho-EGFP variants were subjected to surface immunofluorescence using an anti-Klotho antibody (red) and imaged via immunofluorescence. (**B**) Cells processed as in (**A**) were imaged using higher exposure times to visualize the small amounts of Klotho-EGFP N614Q at the PM. Nuclei were stained with Hoechst 33342 in blue. Single Apotome sections are shown. Arrows indicated PM localization. Scale bar 10 μm. (**C**,**D**) Lysates and supernatants of HEK293T cells stably expressing Klotho-EGFP variants were separated by SDS-PAGE followed by Western blotting using Klotho antibody. (**D**) Quantification of the ratio of mature/total Klotho-EGFP. sKl indicates shed Klotho. n = 4 independent experiments. (**E**,**F**) HEK293 cells stably expressing Klotho-EGFP variants were subjected to surface biotinylation, lysed, precipitated with streptavidin beads, separated on SDS-PAGE, and blotted for Klotho and EEA1 as a cytosolic control protein. * indicates Klotho remnant staining from incomplete stripping. (**F**) Quantification of n = 4 independent experiments from (**E**). The ratio of surface/total of Klotho-EGFP WT was set to 100, and the ratio of surface/total of Klotho-EGFP N614Q related to that. A Student’s *t*-test was used. Data are shown as mean ± standard deviation (SD). * denote levels of significance, **** indicating *p* < 0.0001, ns: non-significant. For full-size blots, see Appendix A.

**Figure 3 cells-13-01743-f003:**
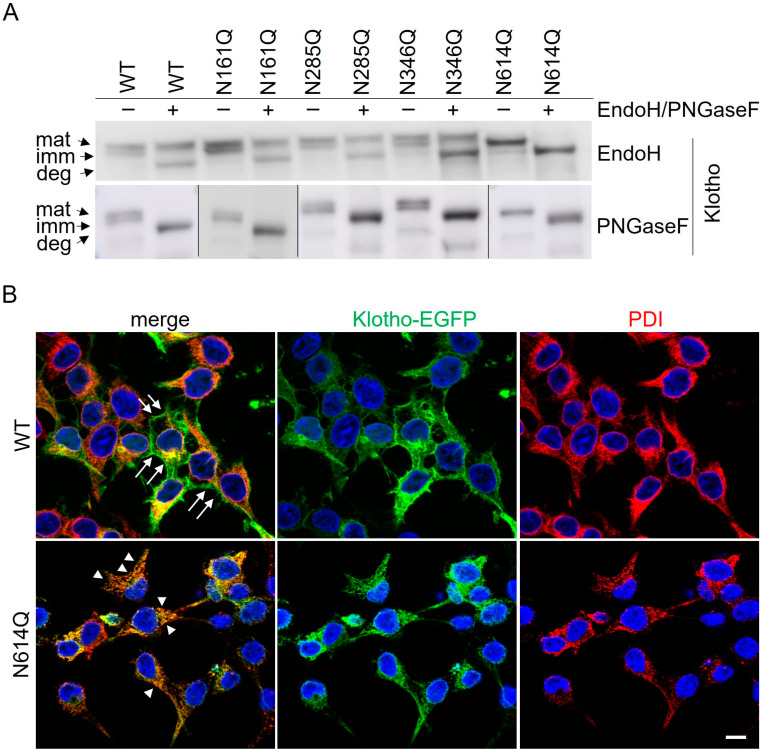
Klotho-EGFP N614Q remains EndoH-sensitive and colocalizes with the ER marker PDI. (**A**) Lysates of HEK293 cells stably expressing Klotho-EGFP WT or glycosylation mutants, as indicated, were subjected to EndoH or PNGaseF digest, separated by SDS-PAGE, blotted, and probed with antibodies against Klotho. mat, mature; imm, immature; deg, deglycosylated. Vertical black lines indicate splicing; full-size blots are shown in Appendix A. (**B**) HEK293 cells stably expressing Klotho-EGFP WT or N614Q were fixed and stained with the ER marker PDI (red). Nuclei were stained with Hoechst 33342 in blue. Arrows depict PM localization, not colocalizing with PDI; arrowheads depict ER localization, colocalizing with PDI. Scale bar 10 μm.

**Figure 4 cells-13-01743-f004:**
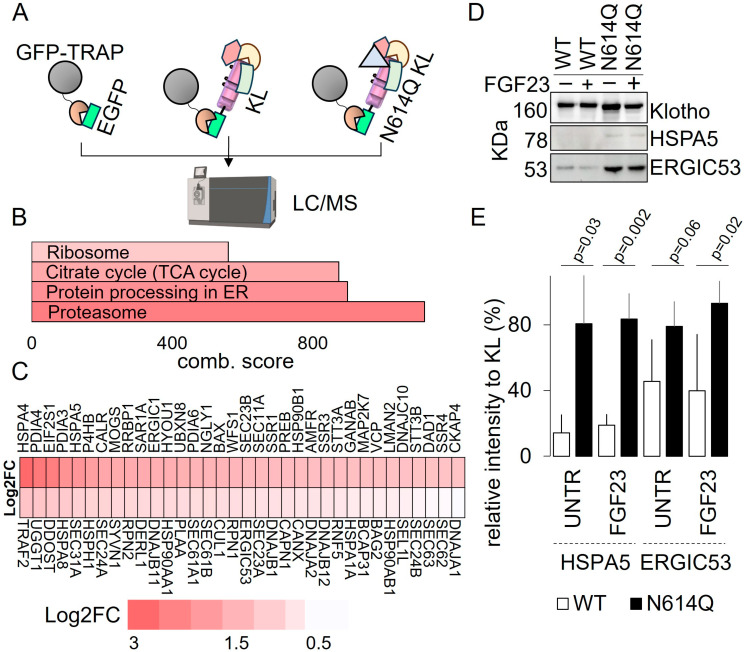
Interaction of Klotho-EGFP N614Q with the proteasome and ER-chaperones. (**A**) Scheme of the GFP-TRAP immunoprecipitation followed by mass spectroscopy (MS). HEK239T cells expressing GFP or Klotho-EGFP variants were used. Created with BioRender.com. (**B**) Bar graph illustrating KEGG pathway analysis of proteins exhibiting increased interaction with Klotho-EGFP N614Q compared to Klotho-EGFP WT (Appendix A). (**C**) Heatmap plot of proteins that exhibit enhanced interaction with Klotho-EGFP N614Q associated with protein processing in the ER (Appendix A). The color code indicates a log2-fold change. (**D**) Western blot analysis of proteins co-IPed with Klotho-EGFP variants from lysates of HEK293T cells stably expressing Klotho-EGFP WT or N614Q after GFP-TRAP IP. Samples were probed with antibodies against Klotho, HSPA5, and ERGIC53. Where indicated, cells were treated with 2.5 nM FGF23 for 5 min. (**E**) Quantification of the interaction of Klotho-EGFP variants with HSPA5 and ERGIC53; displayed is the relative intensity of HSPA5 or ERGIC53 to KL in %. (n = 4 experiments, mean ± SD, two-sided Student’s *t*-test. For full-size blots, see Appendix A.

**Figure 5 cells-13-01743-f005:**
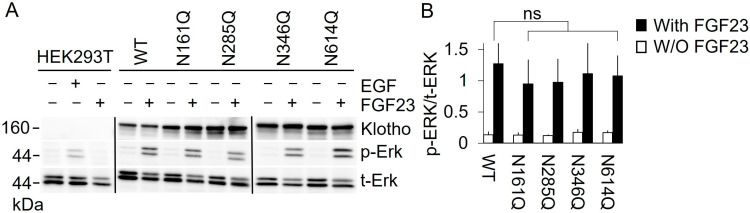
N-glycosylation is not essential for Klotho-FGFR interaction. HEK293T cells stably expressing Klotho-EGFP variants were serum-starved for 16 h and incubated for 5 min with FGF23, followed by lysis and processing for Western blotting with the indicated antibodies. (**B**) Quantification of n = 4 independent experiments, as shown in (**A**). The ratio of the phospho (p)-ERK to total (t)-ERK in the presence and absence of FGF23 is displayed as mean ± SD, with statistical analysis performed using a two-sided Student’s *t*-test (ns: non-significant) Vertical black lines indicate splicing; for full-size blots, see Appendix A.

**Figure 6 cells-13-01743-f006:**
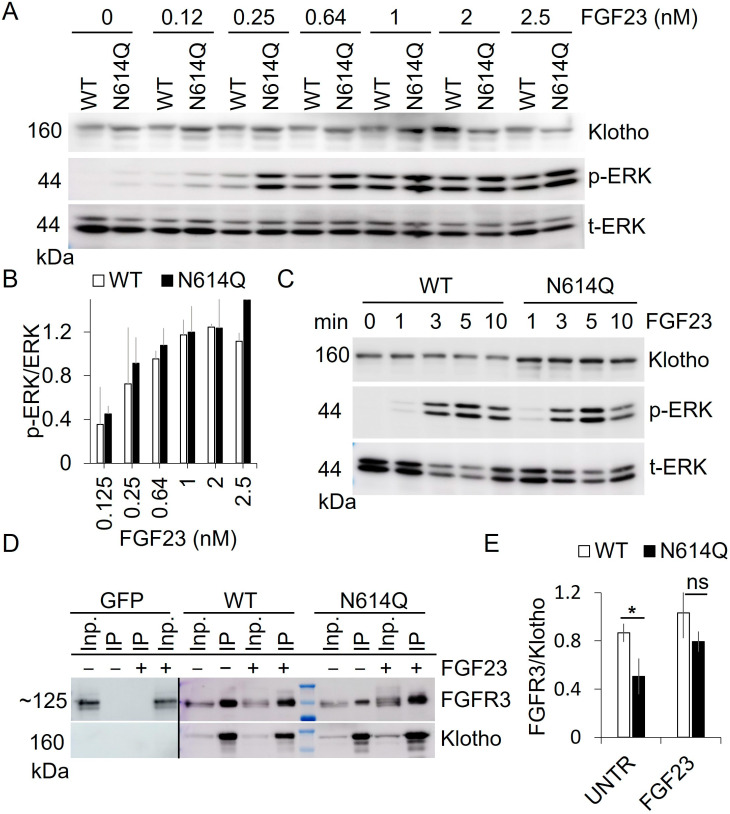
FGF23 does not activate ER-resident Klotho-EGFP-EGFR3 complexes. (**A**,**C**) Western blot of HEK293T cell lysates stably expressing Klotho-EGFP WT or N614Q after treatment with different FGF23 concentrations for 5 min (**A**) or with 2.5 nM FGF23 for the indicated times (**B**), probed with the indicated antibodies. p, phospho;t, total. (**B**) Quantification of n = 4 independent experiments from (**A**). Displayed is the ratio of phospho(p)-ERK normalized to total (t)-ERK. (**C**) A representative blot from n = 2 independent experiments. (**D**) HEK293T cells stably expressing EGFP, Klotho-EGFP WT, or N614Q-treated with 2.5 nM FGF23 for 5 min were subjected to IP with GFP-TRAP. IP lysates (Inp., input) after IP were separated by SDS-PAGE, blotted, and probed with antibodies against FGFR3 or Klotho. (**E**) Quantification of n = 3 independent experiments from (**D**). Displayed is the ratio of FGFR3 to Klotho-EGFP variant. A two-sided Student’s *t*-test was used to assess statistical significance * *p* < 0.05. Error bars indicate SD. Vertical black lines indicate splicing; for full-size blots, see Appendix A.

**Figure 7 cells-13-01743-f007:**
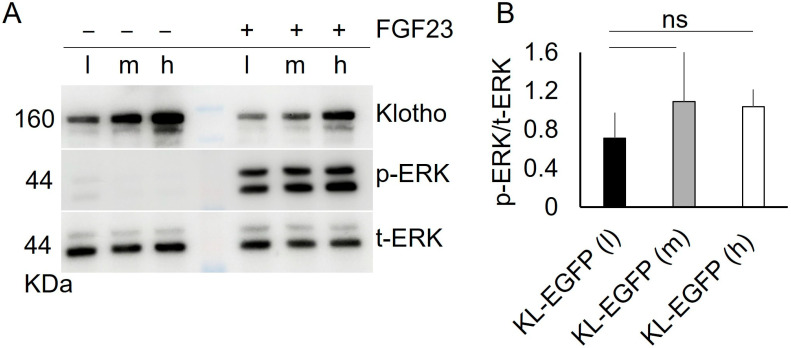
Low amounts of Klotho-EGFP at the plasma membrane are sufficient for ERK activation. (**A**) HEK293T cells stably expressing Klotho-EGFP at low (l), median (m), and high (h) levels were incubated with 2.5 nM FGF23 for 5 min, lysed, separated on SDS-PAGE, and subjected to Western Blotting with the indicated antibodies. (**B**) Quantification of n = 4 independent experiments from A. The displayed ratio is p-ERK/t-ERK. A Two-sided Student’s *t*-test was used to assess statistical significance, (ns: non-significant). Error bars indicate SD. For full-size blots, see Appendix A.

**Figure 8 cells-13-01743-f008:**
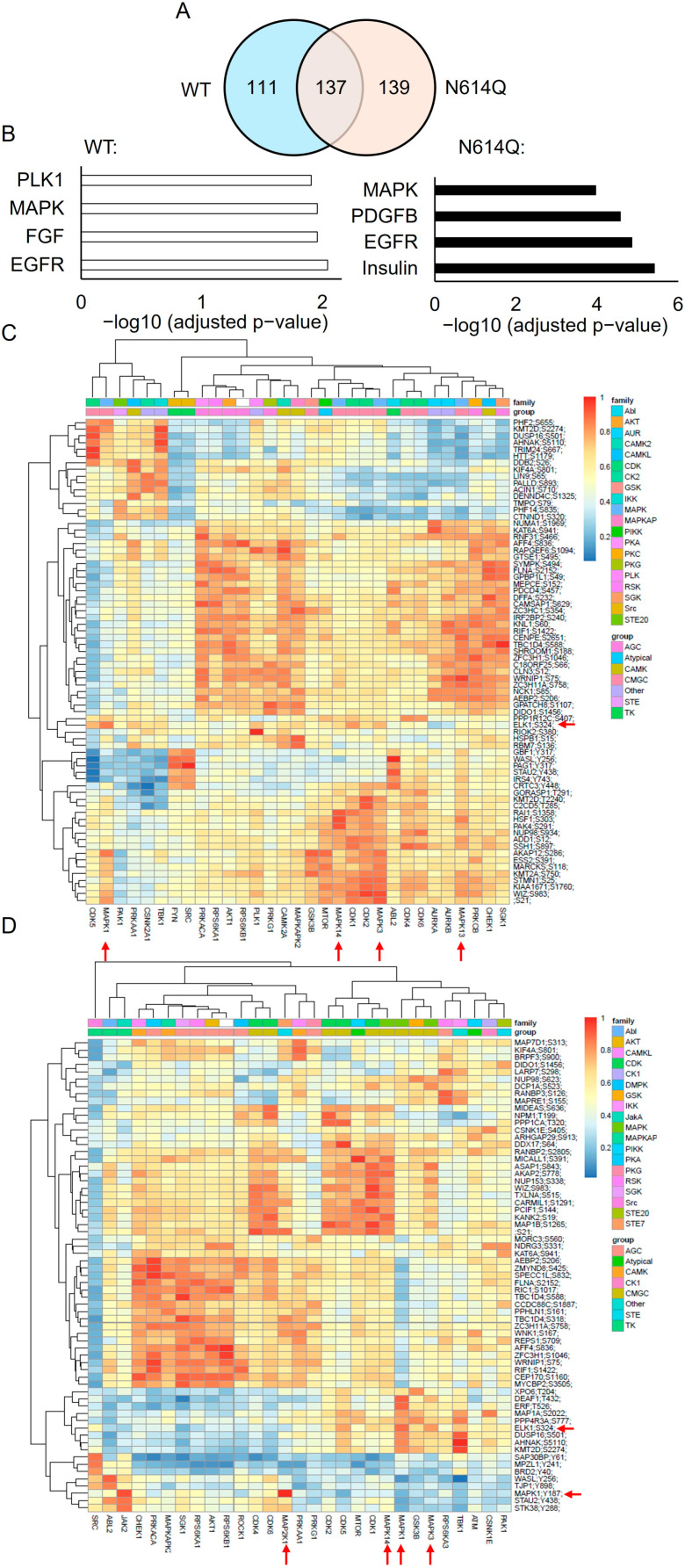
(**A**) Venn diagram illustrating the overlap and distinctions between differentially up-regulated phosphorylated proteins in Klotho-EGFP WT and N614Q mutant after FGF23 treatment. (**B**) Pathway enrichment analysis depicting the differentially up-regulated phosphorylated proteins between Klotho-EGFP WT and N614Q after FGF23 treatment, highlighting the activation of key signaling pathways. (**C**,**D**) Heatmap illustrating the differentially expressed phosphosites and the predicted kinases responsible for their phosphorylation (the rows) in cells expressing Klotho-EGFP WT (**C**) or Klotho-EGFP N614Q (**D**). The red arrows on the right side of each heatmap indicate the presence of the ERK downstream target ELK1 and MAPK1 itself. The arrows at the bottom of the heatmap denote the predicted MAPKs responsible for phosphorylating the phosphosites depicted in the heatmap.

**Table 1 cells-13-01743-t001:** List of antibodies used.

Name	Species	Dilution	Company/Cat No.
Anti-hKlotho	mouse, monoclonal	1:2000 in 5% BSA in TBST for Western Blot	Hölzel Diagnostics, Köln, Germany, SCE-KO604
Phospho-p44/42 MAPK (Erk1/2) (Thr202/Tyr204)	rabbit, monoclonal	1: 5000 in 5% BSA in TBST	Cell signaling technology, Danvers, MA, USA, 4370
p44/42 MAPK (Erk1/2) (L34F12)	mouse, monoclonal	1: 5000 in 5% BSA in TBST	Cell signaling technology, 4696
Anti-Klotho	goat, polyclonal	1:50 in PBSI for immunofluorescence	R&D Systems, AF1819
FGF Receptor 3 (C51F2)	rabbit, monoclonal	1:1000 in I-block	Cell signaling technology, 4574
ERGIC-53 (B-9)	mouse, monoclonal	1:1000 in I-block	Santa Cruz Biotechnology, Dallas, TX, USA, sc-271517
BiP (HSPA5)	rabbit, monoclonal	1:1000 in I-block	Cell Signaling, C50B12
EEA1	mouse, monoclonal	1:1000 in I-block	BD Bioscience, Waltham, MS, USA, 610457
donkey anti-mouse Alexa fluor 555 secondary antibody	donkey	1:1000 in block medium	Invitrogen, Waltham, MS, USA, A31570

**Table 2 cells-13-01743-t002:** Primer sequences used for mutagenesis.

Mutation	Primer	Sequence (5′-3′)	GC%	Tm °C
N161Q	Forward Reverse	CGGGTGCTCCCCCAGGGCACCGCGGGCACTC GAGTGCCCGCGGTGCCCTGGGGGAGCACCCG	83	92
N285Q	Forward Reverse	GTCTGGCATCTCTACCAGACCTCTTTCCGCCCC GGGGCGGAAAGAGGTCTGGTAGAGATGCCAGAC	61	86
N346Q	Forward Reverse	CCAGAGAGTATGAAGAACCAGCTCTCGTCTCTTCTGCC GGCAGAAGAGACGAGAGCTGGTTCTTCATACTCTCTGG	48	81
N614Q	Forward Reverse	CCAGACCCAAGTGCAGCACACGGTTCTGCACTTC GAAGTGCAGAACCGTGTGCTGCACTTGGGTCTGG	56	84

## Data Availability

The original contributions presented in this study are included in the article/Appendix A; further inquiries can be directed to the corresponding author/s. The proteomic datasets generated during and/or analyzed during the current study are available in the MassIve repository, datasets MSV000095613 and MSV000095614. Password: FLI_Reviewer2024.

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
