# Peer review of "Asparagine614 Determines the Transport and Function of the Murine Anti-Aging Protein Klotho"

_cells, 2024, doi:10.3390/cells13201743_

Round 1
Reviewer 1 Report
Comments and Suggestions for Authors
Klotho is an anti-aging protein primarily expressed in renal tubule cells. It is an N-glycosylated protein
that exits in a soluble, membrane-bound form. In this study, the authors investigated the role of
N-glycosylation in the processing and trafficking of Klotho to the cell surface. Their main conclusion
is that N-glycosylation determines Klotho transport and function. Overall, the study is technically well
executed and the article has the potential to be very interesting. However, several aspects need to be
addressed adequately as outlined below:
1: Fig2 A is nice and the images provide evidence indeed that, in contrast to N161Q, N285Q, and N346Q, N614Q is trapped in the ER and therefore is not detectable at the cell surface. However, Fig.2 B (obtained using higher exposure times) is not convincing, and the authors are too affirmative by claiming that very low amounts of N161Q are expressed at the cell surface. In these experiments, the authors used an anti-klotho antibody. Given that klotho is endogenously expressed in HEK cells [1, 2], one cannot discount the possibility that the weak signal detected at the cell surface in Fig.2 B, is simply due to endogenous klotho, and not to N161Q. Further investigations using a specific plasma membrane marker or cell surface biotinylation followed by co-immunolocalization experiments are required to check whether EGFP-N614Q is capable of reaching the cell surface. Otherwise, one cannot draw any solid conclusion based only on Fig. 2B.
2: The lower panel of Fig. 2C clearly shows that N614Q is not secreted, which is consistent with ER retention of the protein. However, I am concerned about the author's interpretation of the data presented in the upper panel (WB on total cell lysate) in claiming that “N-glycosylation” is required for cell surface expression and secretion of Klotho. Given that N-glycosylation at one site can only account for 4–20 kDa, a slight change in the molecular weight of the immature and mature form of klotho is expected. This appears to be the case for N161Q, N285Q, and N346Q (Fig2 B, upper panel), and this may also serve as an indication that these three N-glycosylation sites are functional. Can the authors confirm this? What is the expected MW of the different forms of the proteins? It would be very helpful to add the MW next to each band and comment on the presence (or not) of a small shift after each mutation. The “unexpected” major loss of the fully-glycosylated form of klotho after mutating only N614Q is interesting. However, I am not sure it is appropriate to conclude that this is attributable to the N-glycosylation process per se. N614Q most likely behaves as a simple folding mutant trapped by ER quality control. I agree with the authors that the data strongly suggest that N161 is important for the correct folding of Klotho, but this does not mean that N-glycosylation, per se, is necessary for the processing and transport of the protein to the cell surface.
3: Fig 5 and 6: The data may suggest that low amounts of “Wild Type” klotho are sufficient for ERK-activation but the issue here is with N614Q and not WT EGFP KL. Again, these data are interesting but more efforts should be devoted to determining whether N614Q is indeed expressed at the cell surface. Consequently, further analysis, for instance, using western blotting of biotinylated cell surface protein, is required to accurately determine the amount of EGFP-N614Q (compared to that of WT klotho) expressed at the cell surface. These data are crucial for the authors to be more convincing. Otherwise, no solid conclusions can be drawn at this stage on this issue.
4: The title of the manuscript is misleading given that mutation of Klotho's N-glycosylation sites did not affect its function. Moreover, even impaired surface transport of Klotho-EGFP N614Q does not block activation of the FGFR signaling pathway.. Hence, the authors provided no evidence that N-glycosylation determines Klotho function. Furthermore, mutation of the other three glycosylation sites (N161Q, N285Q, and N346Q) had not affect on Klotho transport to the cell surface. The authors must therefore be more specific in defining a title that accurately reflects the content of the manuscript.
References:
1. Zhang, H., et al., Klotho is a target gene of PPAR-γ. Kidney International, 2008. 74(6): p. 732-739.
2. Li, Y., et al., Klotho is regulated by transcription factor Sp1 in renal tubular epithelial cells. BMC Molecular and Cell Biology, 2020. 21(1): p. 45.
Author Response
Klotho is an anti-aging protein primarily expressed in renal tubule cells. It is an N-glycosylated
protein that exits in a soluble, membrane-bound form. In this study, the authors investigated
the role of N-glycosylation in the processing and trafficking of Klotho to the cell surface. Their
main conclusion is that N-glycosylation determines Klotho transport and function. Overall, the
study is technically well executed and the article has the potential to be very interesting.
However, several aspects need to be
addressed adequately as outlined below:
1: Fig2 A is nice and the images provide evidence indeed that, in contrast to N161Q, N285Q,
and N346Q, N614Q is trapped in the ER and therefore is not detectable at the cell surface.
However, Fig.2 B (obtained using higher exposure times) is not convincing, and the authors
are too affirmative by claiming that very low amounts of N161Q are expressed at the cell
surface. In these experiments, the authors used an anti-klotho antibody. Given that klotho is
endogenously expressed in HEK cells [1, 2], one cannot discount the possibility that the weak
signal detected at the cell surface in Fig.2 B, is simply due to endogenous klotho, and not to
N161Q. Further investigations using a specific plasma membrane marker or cell surface
biotinylation followed by co-immunolocalization experiments are required to check whether
EGFP-N614Q is capable of reaching the cell surface. Otherwise, one cannot draw any solid
conclusion based only on Fig. 2B.
To further prove that it is not the endogenous surface Klotho we detect in N614Q expressing
cells we redid the immunofluorescence experiment and compared surface staining of wild-type
HEK293 and N614Q expressing cells in identical settings. In the revised Fig. 2B no signal is
visible in untransfected HEK293 cells compared to the weak signal in N614Q expressing cells,
demonstrating it is indeed the overexpressed, Klotho-EGFP N614Q that we detect. In addition,
we conducted cell surface biotinylation to get a quantitative comparison of the cell surface
localization of Klotho-EGFP wt and N614Q variant, see revised Fig. 2E,F.
We would also like to mention that the addition of FGF23 to untransfected HEK293 cells does
not elicit ERK phosphorylation (in contrast to N614Q-expressing cells! Fig. 4), strongly
suggesting that the endogenous Klotho is not interfering with our assays, neither in the
immunofluorescence nor in the signaling experiments.
2: The lower panel of Fig. 2C clearly shows that N614Q is not secreted, which is consistent
with ER retention of the protein. However, I am concerned about the author's interpretation of
the data presented in the upper panel (WB on total cell lysate) in claiming that “N-glycosylation”
is required for cell surface expression and secretion of Klotho. Given that N-glycosylation at
one site can only account for 4–20 kDa, a slight change in the molecular weight of the immature
and mature form of klotho is expected. This appears to be the case for N161Q, N285Q, and
N346Q (Fig2 B, upper panel), and this may also serve as an indication that these three Nglycosylation
sites are functional. Can the authors confirm this? What is the expected MW of
the different forms of the proteins? It would be very helpful to add the MW next to each band
and comment on the presence (or not) of a small shift after each mutation. The “unexpected”
major loss of the fully-glycosylated form of klotho after mutating only N614Q is interesting.
However, I am not sure it is appropriate to conclude that this is attributable to the Nglycosylation
process per se. N614Q most likely behaves as a simple folding mutant trapped
by ER quality control. I agree with the authors that the data strongly suggest that N161 is
important for the correct folding of Klotho, but this does not mean that N-glycosylation, per se,
is necessary for the processing and transport of the protein to the cell surface.
You are right, we cannot prove that it is the missing N-glycosylation at N614 and not the change
of N614 to Q in the primary sequence that causes the misfolding and ER-retention. We discuss
now the alternative that N614Q is resulting in misfolding independent of the glycosylation at
that site and changed the title and the relevant parts in results and discussion.
We confirm the reviewer´s observation that N161Q, N285Q, N346Q but also N614Q results in
slight changes in molecular weight, in agreement with a 4-20 kDa shift. N614Q in Fig. 2 might
appear to lead to a slightly bigger shift than the other mutants, but if you compare the shifts in
Fig. 3 (previously Fig. S1) in both Endo H and PNGaseF digests there are no indications for a
larger shift. This suggests that in all cases the mutated asparagines are functional Nglycosylation
sites and that in all mutants only the N-glycosylation at that site is affected. We
added an estimated MW based on the MW markers we used, but we were not able to robustly
predict the exact MW of each band, since, as also the reviewer points out, N-glycosylation at
single sites can vary considerably, plus Klotho is also O-glycosylated and may undergo
additional posttranslational modifications.
3: Fig 5 and 6: The data may suggest that low amounts of “Wild Type” klotho are sufficient for
ERK-activation but the issue here is with N614Q and not WT EGFP KL. Again, these data are
interesting but more efforts should be devoted to determining whether N614Q is indeed
expressed at the cell surface. Consequently, further analysis, for instance, using western
blotting of biotinylated cell surface protein, is required to accurately determine the amount of
EGFP-N614Q (compared to that of WT klotho) expressed at the cell surface. These data are
crucial for the authors to be more convincing. Otherwise, no solid conclusions can be drawn
at this stage on this issue.
As requested we did surface biotinylation to show that N614Q is indeed at the cell surface,
albeit in low amounts. See revised Fig. 2E,F and comments to your comment #1.
4: The title of the manuscript is misleading given that mutation of Klotho's N-glycosylation sites
did not affect its function. Moreover, even impaired surface transport of Klotho-EGFP N614Q
does not block activation of the FGFR signaling pathway.. Hence, the authors provided no
evidence that N-glycosylation determines Klotho function. Furthermore, mutation of the other
three glycosylation sites (N161Q, N285Q, and N346Q) had not affect on Klotho transport to
the cell surface. The authors must therefore be more specific in defining a title that accurately
reflects the content of the manuscript.
We changed the title. “Asparagine614 determines transport and function of the murine antiaging
protein Klotho” now hopefully better reflects the content of the manuscript.
Reviewer 2 Report
Comments and Suggestions for Authors
The work by Fanaei-Kahrani and Kaether presents the study of the role of glycosylation, and more specifically, of site specific glycosylation as regulator of Klotho glycoprotein presentation on the plasma membrane and on how this glycosylation influences its transport from the ER to the Golgi, and finally to the cellular membrane, as well as on its interaction with proteasome-related protein. The work is very interesting and the data are very clearly presented. During the last years, the number of works showing the role of protein glycosylation as regulator of glycoprotein trafficking, interactions, downstream signaling etc etc, is significantly increasing. This demonstrate the high interest from the scientific community in understanding the role of glycosylation. This work is a very nice contribution to the field. Few comments:
-I suggest to clearly indicate in the title that the analysed Klotho glycoprotein is a murine protein. This is because, as the authors say, the human type presents 8 N glycosylation site. This means that the conclusions made by the authors on the dramatic influence of the N614Q in the murine form, that has only 4 N-glycosylation, may not be so relevant in the human Klotho.
-In the result section, 3.1, the authors claim that the confirm the N-glycosylation of Klotho, the protein was treated with PNGaseF or EndoH. Please specify whether the protein was denatured or not. This is important to confirm that all the glycosylation sites were equally accessible to the enzymes.
-In Figure 1A, the Western Blotting of the Klotho protein after Endo H treatment is very weak. Please explain why?
-Suppl. Fig.S1 demonstrate that Klotho-EGFP N614Q is mostly retained in the ER. I consider this figure very important and should be moved to the main text.
-Under the structural point of view, the authors suggest that the N614 is essential for proper Klotho folding. I suggest to use any structural technique to demonstrate that the N614Q is actually unfolded. This could include circular dichroism, NMR or others.
Author Response
The work by Fanaei-Kahrani and Kaether presents the study of the role of glycosylation, and
more specifically, of site specific glycosylation as regulator of Klotho glycoprotein presentation
on the plasma membrane and on how this glycosylation influences its transport from the ER
to the Golgi, and finally to the cellular membrane, as well as on its interaction with proteasomerelated
protein. The work is very interesting and the data are very clearly presented. During
the last years, the number of works showing the role of protein glycosylation as regulator of
glycoprotein trafficking, interactions, downstream signaling etc etc, is significantly increasing.
This demonstrate the high interest from the scientific community in understanding the role of
glycosylation. This work is a very nice contribution to the field. Few comments:
-I suggest to clearly indicate in the title that the analysed Klotho glycoprotein is a murine
protein. This is because, as the authors say, the human type presents 8 N glycosylation site.
This means that the conclusions made by the authors on the dramatic influence of the N614Q
in the murine form, that has only 4 N-glycosylation, may not be so relevant in the human Klotho.
We were not clear in explaining our rationale. Actually both mouse and human Klotho have 8
N-glycosylation sites, but we decided to analyze only four, two previously suggested to reduce
secretion, two with normal secretion in a soluble human Klotho-construct. Nevertheless, we
agree that is important to indicate in the title that we used murine Klotho. We changed the title
accordingly and also explained more precisely in the introduction what we did.
-In the result section, 3.1, the authors claim that the confirm the N-glycosylation of Klotho, the
protein was treated with PNGaseF or EndoH. Please specify whether the protein was
denatured or not. This is important to confirm that all the glycosylation sites were equally
accessible to the enzymes.
Thanks for pointing this out, we realize that our method section was incomplete, it lacked any
information about the deglycosylation. We corrected that, and yes, there is a denaturing step.
-In Figure 1A, the Western Blotting of the Klotho protein after Endo H treatment is very weak.
Please explain why?
This is just experimental variation, we have other replicates where there is more Klotho after
EndoH treatment, one example is shown in extra figure E1.
-Suppl. Fig.S1 demonstrate that Klotho-EGFP N614Q is mostly retained in the ER. I consider
this figure very important and should be moved to the main text.
We followed the suggestion and moved Fig. S1 to the main text, now Fig. 3.
-Under the structural point of view, the authors suggest that the N614 is essential for proper
Klotho folding. I suggest to use any structural technique to demonstrate that the N614Q is
actually unfolded. This could include circular dichroism, NMR or others.
Structural techniques would indeed be very helpful to show that the N614Q mutant is indeed
unfolded. However, this would necessitate recombinant expression of probably only the
luminal part (the full-length Klotho is a transmembrane protein) and some functional testing of
the recombinant proteins. We do not have experience in that direction, nor do we have access
to NMR or circular dichroism, therefore we think such an analysis is beyond the scope of the
current manuscript. Strong evidence that the N614Q mutant is indeed unfolded comes from
the co-immunoprecipitation of BiP (HSPA5), the paradigmatic binding protein of unfolded
proteins, and of the detection of BiP and the other chaperones in the MS-data (Fig. 4,
previously Fig. 3).

Reviewer 3 Report
Comments and Suggestions for Authors This study examined the effects of N-glycosylation determines transport and function of the anti-aging protein Klotho. Although these studies were intresting and usefull for providing theoretical references for the anti-aging protein Klotho. However, there are still be some information or description should be added or changed in this paper. 1. Abstract: authors can add more key findings or highlightings in this study. 2. FACS and other abbreviations should add their whole name in this paper. 3. In Fig. 2, authors should add arrows to mark characters in this study. 4. Authors should add more information about these sample replicate numbers in the methods. 5. Authors should add more findings in the Conclusions, such as interacting proteins.
Comments on the Quality of English Language
the Quality of English Language is good.
Author Response
This study examined the effects of N-glycosylation determines transport and function of the
anti-aging protein Klotho. Although these studies were intresting and usefull for providing
theoretical references for the anti-aging protein Klotho. However, there are still be some
information or description should be added or changed in this paper.
1. Abstract: authors can add more key findings or highlightings in this study.
We changed the abstract accordingly and specified our findings.
2. FACS and other abbreviations should add their whole name in this paper.
We thoroughly went through the manuscript and explained abbreviations like FGFR, FACS,
MAPK etc.
3. In Fig. 2, authors should add arrows to mark characters in this study.
We added some arrows in A) to highlight PM-staining
4.Authors should add more information about these sample replicate numbers in the methods.
We added the requested information
5. Authors should add more findings in the Conclusions, such as interacting proteins.
We added more findings to the conclusions.
Round 2
Reviewer 1 Report
Comments and Suggestions for Authors
The authors have adequately addressed all of my concerns. I consider the manuscript acceptable for publication in its present form.